# A Review on Gold Nanotriangles: Synthesis, Self-Assembly and Their Applications

**DOI:** 10.3390/molecules27248766

**Published:** 2022-12-10

**Authors:** Xiaoxi Yu, Zhengkang Wang, Handan Cui, Xiaofei Wu, Wenjing Chai, Jinjian Wei, Yuqin Chen, Zhide Zhang

**Affiliations:** College of Chemistry, Chemical Engineering and Materials Science, Shandong Normal University, Jinan 250014, China

**Keywords:** gold nanotriangles, synthesis, self-assembly, plasmonic applications

## Abstract

Gold nanoparticles (AuNPs) with interesting optical properties have attracted much attention in recent years. The synthesis and plasmonic properties of AuNPs with a controllable size and shape have been extensively investigated. Among these AuNPs, gold nanotriangles (AuNTs) exhibited unique optical and plasmonic properties due to their special triangular anisotropy. Indeed, AuNTs showed promising applications in optoelectronics, optical sensing, imaging and other fields. However, only few reviews about these applications have been reported. Herein, we comprehensively reviewed the synthesis and self-assembly of AuNTs and their applications in recent years. The preparation protocols of AuNTs are mainly categorized into chemical synthesis, biosynthesis and physical-stimulus-induced synthesis. The comparison between the advantages and disadvantages of various synthetic strategies are discussed. Furthermore, the specific surface modification of AuNTs and their self-assembly into different dimensional nano- or microstructures by various interparticle interactions are introduced. Based on the unique physical properties of AuNTs and their assemblies, the applications towards chemical biology and sensing were developed. Finally, the future development of AuNTs is prospected.

## 1. Introduction

Plasmonic metallic nanoparticles such as gold [1,2,3,4], silver [5,6,7,8] and copper [9,10] exhibit interesting optical properties, which originate from the collective and coherent oscillations of conductive electrons under the irradiation of light [11]. The resulting resonance band from localized surface plasmon resonance (LSPR) is largely influenced by the composition, shape and size of nanoparticles [12,13]. Several comprehensive reviews introduced the synthesis and plasmonic properties of the metallic nanoparticles with different shapes such as spheres, nanocages, rods, nanomatryoshkas, stars, dendrites, icosahedrons, dumbbells and dimmers [13,14,15,16,17,18,19,20,21,22,23,24]. Herein, we briefly summarized the classifications, plasmonic bands and synthetic methods of different nanoparticles in Table 1. Compared with silver and copper nanoparticles, gold nanoparticles (AuNPs) with good biocompatibility and stability have attracted much attention in recent years due to their potential applications in sensing, optical materials, catalysis and biological fields [25,26,27,28,29,30]. Different from the isotropic nanoparticles, anisotropic nanoparticles such as nanorods and nanostars exhibited disparate plasmonic resonance modes, thus resulting in their unique optical characteristics [30,31,32]. Gold nanorods showed transversal and longitudinal resonances, resulting in two typical LSPR bands [33,34,35]. The plasmonic peaks were largely influenced by the length, width and aspect ratio between them. Gold nanostars are another complex nanostructures composed of inner core and outer spikes [36,37]. The length of the spikes dramatically influenced the absorption by nanostars. With increasing spike lengths, the absorption became red-shifted.

Among the various shaped AuNPs, gold nanotriangles (AuNTs) with a unique triangular anisotropy exhibited special optical properties, in which the three sharp vertices exhibited a significant hot spot enhancement effect. AuNTs also showed a wide and controllable LSPR by changing their sizes compared with other shaped AuNPs (Table 1). The LSPR modes in AuNTs were proven to locate at the corners, edges and two faces (up/down) depending on the wavelength of the incident light. As a result, the LSPR spectrum of AuNTs showed two plasmonic peaks with a typical band above 600 nm and another weak shoulder band around 530 nm [38,39]. The LSPR band of AuNTs could be well tuned by controlling their edge length, thickness and ratio. Another merit of AuNTs is that their highly-curved corners ensure a large electric field enhancement. Electron energy-loss spectroscopy (EELS) experimental data of AuNTs supported that the tips provided a high electromagnetic field enhancement from a dipolar plasmon mode, while other modes at higher energies originated from the highest enhancement at the edges and at the center of the AuNTs [40]. The boundary-element method (BEM) simulation supported the existence of the in-plane dipole and quadrupole plasmon modes in AuNT and the EELS mapping at the corresponding energies [13,40]. The energy of the plasmon mode at the corner was lower than those at the edge and face. With the increase in AuNT thickness, the energy of corner decreased, while the edge and face increased. This energy shift was caused by an increase in the free electron density in the AuNT [39]. Theoretical calculations indicated that the EELS and cathodoluminescence were closely related to the optical extinction and scattering of AuNTs, respectively. The dipolar mode could induce scattering blue-shifts in connection with its corresponding extinction. The magnitude of the extinction and scattering increased with the increasing dissipation level [40]. During the past decades, many efforts were devoted to the synthetic exploration of AuNTs with high-quality and obtained yields. Various synthetic approaches of AuNTs based on the chemical synthesis, biosynthesis and physical-stimulus-induced synthesis have been developed [41,42,43,44,45]. As a result, the self-assembly of AuNTs with specific surface modifications into different dimensional nano- or microscale structures for their potential applications have been also emerged [46,47]. This review focused on the research progress of AuNTs from the aspects of their synthesis, self-assembly by surface modification and applications (Figure 1).

## 2. Synthesis of AuNTs

### 2.1. Chemical Synthesis

#### 2.1.1. Seed-Mediated Growth Method

The preparation of AuNTs with a controllable size started began by using the thermal method [51]. In this preparation, the mixture of ethylene glycol, HAuCl_4_ and PVP was quickly injected into boiling ethylene glycol. Ethylene glycol was not only used as a solvent, but also for reducing HAuCl_4_. Poly(vinylpyrrolidone) (PVP) could stabilize the particles and control the shape of the particles through the selective interaction with surface planes of the gold nanocrystals [52]. However, this thermal method for preparation of AuNTs was limited to the application for organic solvents, but not to aqueous solutions, thus hindering its practical application. Therefore, the seed-mediated synthesis of AuNTs was developed via a three-step growth process in water by the Mirkin group [53]. However, a large number of spherical nanoparticles existed in this method, resulting in the low obtained yield of AuNTs. It was demonstrated that the Au seed served as the nucleus for epitaxial growth in the seed-mediated method. Thus, the initial seed structure is essential in determining the final nanoparticle morphology. Some of the literature has shown that halide anions as shape-directing agents played an important role in the formation of the Au seed [54,55,56,57]. The combination of halide anions could promote preferential growth in specific crystallographic directions, resulting in the formation of different nanostructures. AuNTs were efficiently formed by using the combination of iodide and chloride anions or iodide and bromide anions in specific ratios. Iodide anions with the largest strength of absorption and the highest degree of specificity could preferentially adsorb onto Au {111} planes to form AuNTs as the dominant morphology. Furthermore, the combination of iodide and bromide anions showed a higher efficiency to stabilize the Au {111} facet than that of the iodide and chloride anion pairs to produce AuNTs.

Recently, the Liz-Marzán group developed a new seed-mediated growth method to prepare AuNTs with high quality [58]. This method generally included three procedures (Figure 2a): (1) Synthesis of small spherical initial seeds using cetyltrimethylammonium chloride (CTAC) as the surfactant—HAuCl_4_ was reduced to gold atoms by using NaBH_4_ as the reducing agent; (2) Fast seed addition process—two growth solutions were prepared, with the addition of seed to the growth solution 1 for the formation of larger nanoparticles with triangular and spherical shapes. The above solution was rapidly added to the growth solution 2 for the formation of AuNTs; (3) Purification: the purification was achieved by the depletion force. It is worth mentioning that this seed-mediated growth method separated the nucleation and growth processes step by step, resulting in the high conversion into AuNTs. The size of AuNTs could be controlled by varying the volume of the seed solution and the concentration of iodide. However, this synthetic method had obvious disadvantages; key step relied on the rapid addition technology, and the reproducibility and shape yield of AuNTs could be affected by inexperienced researchers.

To achieve the practical performance of AuNTs with good reproducibility, it was important to control the overgrowth of initial seeds into a stable state of intermediate seeds. The overgrowth of the initial seeds using 1/3 of the molar content of Au compared with the previous report could obtain stable intermediate seeds for the efficient synthesis of AuNTs without the “fast addition” process [47]. This seed solution could be stable for more than 6 days at a temperature of 4 °C. On the other hand, the combination of CTAC and a new additive such as PVP as a stabilizer could synthesize monodisperse AuNTs of different sizes with enhanced stability [59]. Compared to adding the single surfactant CTAC, the more stable AuNTs were obtained by adding nanomoles of PVP during the growth process (Figure 2a). Experimental data showed that PVP chains interacted with the surface of AuNTs through the oxygen of a carbonyl group. Further, PVP could interact more favorably with Au {111} facets, which decreased the growth rate along the {111} direction while enhancing the anisotropic growth rate along the {100} direction. The reducing reagents with different reducing ability were also important for the preparation of AuNTs. 3-Butenoic acid (3BA) with moderate reduction efficiency could reduce HAuCl_4_ in the presence of the capping agent benzyldimethylammonium chloride (BDAC) for the controllable synthesis of AuNTs [60]. 3BA was unable to reduce Au^3+^ to Au^0^ at room temperature, whereas when the temperature increased above 60 °C, the reducing ability of 3BA was improved and reduced HAuCl_4_ to promote the nucleation process. By changing the solution from acidulous to alkalescence, the synthetic process gradually changed from thermodynamic control to kinetic control, which led to the decline of the shape yield. This indicated that the synthesis of AuNTs could be precisely controlled by adjusting the temperature or pH. Furthermore, the critical concentration of BDAC to form micelles was lower than that of CTAB. The BDAC solution had a larger depletion potential than the CTAB solution under the same concentration to achieve efficient purification.

#### 2.1.2. Molecular Vesicle Phase with Alternating Polyampholyte

The mixed phospholipid-based vesicles were used to synthesize the anisotropic gold nanoparticles since the vesicular template could direct the anisotropic growth in a characteristic way based on the kinetic control [62]. The addition of strong alternating polyampholytes such as PalPhBisCarb could also facilitate AuNT formation. Kinetic studies showed that the yield and edge length of AuNTs were largely influenced by varying the dose of polyampholyte and the incubation temperature. The zeta potential studies suggested that the temperature-dependent absorption of polyampholytes on {111} faces induced a symmetry breaking. This resulted in the prevention of the kinetically-controlled vertical growth and the preferred lateral growth of AuNTs. The preparation of AuNTs could be achieved in a green chemical vesicle phase formed by phospholipids in water without an external reducing reagent [61,63]. The AuNTs were purified by repeated centrifugation and by adding surfactant micelles to the mixture for flocculation (Figure 2b).

#### 2.1.3. Photocatalytic Reduction

The photocatalytic reduction of gold salt to form AuNTs in water could be achieved by adding the photocatalyst Tin(IV) porphyrin and the electron donor triethanolamine in the presence of cetyltrimethylammonium bromide (CTAB) [64]. The average edge length of AuNTs could be controlled by changing the concentration of Tin(IV) porphyrin or the concentration of CTAB (Figure 2c). Generally, the average size of AuNTs decreased with the increasing concentration of CTAB, since the nanoparticle growth was hindered by the surface CTAB. Furthermore, the photocatalysts could independently control the kinetics of gold reduction, and other experimental factors such as the concentration of the capping agents and pH also influenced the AuNT formation. This method could produce uniform AuNTs with high selectivity and yield.

#### 2.1.4. Nucleus-Free Method

The one-pot nucleus-free oxidation etching process to prepare monodisperse AuNTs without using a purification process could be achieved by using triiodide (I_3_^−^) with oxidative properties [65]. In this method, the iodide ions had dual functions: selectively binding to the Au{111} facets to promote the formation of AuNTs and the formation of I_3_^−^ to selectively remove unstable other shape nanoparticle impurities at an early stage by oxidative etching (Figure 2d). This method was likely to be extended to other anisotropic gold nanostructures, which could not only improve the yield and uniformity of existing nanostructures, but also make practical applications more economical. Th seed-mediated growth approach to prepare AuNTs used the cytotoxic surfactants CTAB or CTAC, and their toxicity limited the direct contact of AuNTs with healthy tissues [66]. CTAB- or CTA-free preparations of AuNTs typically involved the use of mild inorganic reducing agents (Na_2_S_2_O_3_ and KI) in combination with a directing agent such as the zwitterionic thiol-containing molecule glutathione (GSH). Since GSH at various pH values promoted interparticle interactions, the AuNTs could be purified through sediment under a selected pH without using the repeated purification based on depletion force. However, it was difficult to control the size of AuNTs.

### 2.2. Biosynthesis

The synthesis of AuNTs using chemical methods is time consuming, costly and harmful to the environment. In recent years, the biosynthesis of AuNTs from the extracts of plants, fungi and bacteria has attracted great interest because of the low cost, high reproducibility, environmental friendliness and easy availability of raw materials. Sastry and coworkers reported the biosynthesis of single-crystal AuNTs by the reduction of chloroaurate ions (AuCl_4_^−^) using extracts of plant lemongrass (Cymbopogon Flexuosus) at room temperature [67,68]. The ketone/aldehyde fraction of the extract could modify the surface of small nanoparticles to a “liquid-like” property. These nanoparticles were anisotropically sintered to each other at room temperature to form AuNTs. The aloe vera leaf extract served as the reducing agent for the reaction of aqueous HAuCl_4_ to synthesize AuNTs [69]. Experimental data showed that the multiple twinned particles gradually transformed into flat triangular nanostructures. The slow reduction of gold ions in the aqueous phase and the shape-directing effect of the aloe vera extract components were responsible for the formation of AuNTs (Figure 3a). The shape transformation of nanoparticles from spheres and triangles could be controlled by varying the percentage of extract in the reaction medium [70]. The specific Streptomyces isolated from rice could also synthesize AuNTs via a fast, simple and environmentally-friendly process [71]. Among the investigated 15 Streptomyces isolates from the paddy fields, isolate No. 5 showed high extracellular biosynthetic AuNTs activity after treating it with HAuCl_4_ aqueous solution.

In addition to using extracts from plants for AuNT synthesis, the endophytic fungus Aspergillus clavatus isolated from surface-sterilized stem tissue of Azadirachta indica A.as the reduction reagent could also synthesize AuNTs by a single-step “green biosynthesis” [72]. When Aspergillus clavatus was incubated with an aqueous solution of chloroaurate ions, a mixture of various intracellular AuNTs was produced, in which the AuNTs were 20–35 nm in size and displayed high aspect ratios. However, the quality of the biological raw materials might not be uniform in various regions or countries. Furthermore, the extraction conditions such as the extraction solvent and temperature largely influenced the purity of the active target substances. These factors could affect the efficiency and morphology control of AuNT synthesis.

### 2.3. Physical-Stimulus-Induced Synthesis

#### 2.3.1. Electric-Field-Assisted Growth

Electric fields could induce the directed motion of surface charged nanoparticles to the conductive substrate to form highly-ordered AuNT arrays [74]. This method included a seed-mediated growth process and electrochemical deposition of metal particles on the indium tin oxide (ITO)-coated conductive glass surface (Figure 3b). Ascorbic acid reduced HAuCl_4_ to gold seeds, which were attached on the surface of the substrate by immersing it in a seed solution and washing it with water. The electric field could induce the anisotropic growth of nanoparticles into AuNTs on the substrate in the growth solution. This electric potential might enhance the preferential absorption of a positively-charged CTAB stabilizer to specific crystallographic planes. Potentially-assisted chemical growth was used to synthesize highly uniform equilateral AuNTs arrays on the ITO surface, which showed high NIR absorption and strong surface-enhanced Raman activity. These well-ordered AuNTs might have potential in optoelectronic devices and biosensors.

#### 2.3.2. Laser Irradiation Induced AuNT Formation

Levis et al. reported the use of laser-generated gold clusters to create AuNTs with near-quantitative conversion [73]. The Au clusters were partially produced through the short-time (5 s) irradiation of aqueous KAuCl_4_ by femtosecond laser (Figure 3c). The slow addition of H_2_O_2_ reduced the surplus aqueous KAuCl_4_ into AuNTs in the presence of the above seeds without adding capping reagents such as CTAB. The mechanism of AuNT formation might include three key steps: Au seeds formation by strong-filed laser processing; seed-particle-oriented attachment together at the corrected crystallographic alignment; and crystalline nanoplate formation via recrystallization of nanoplates by the continued attachment of seed particles at their edges. The effect of the irradiation of aqueous KAuCl_4_ by femtosecond laser followed by post-irradiation of H_2_O_2_ for reduction showed that the irradiation time of laser processing was important for AuNT formation [75]. Longer laser irradiation (240 s) induced a full reduction of Au (III) to spherical particles with an average size of 11.4 nm without the formation of AuNTs, while short irradiation resulted in the formation of smaller AuNP seeds of 5.7 nm together with the unreacted aqueous KAuCl_4_. The post-irradiation reduction reaction within a short irradiation time induced the formation of a large number of AuNTs. The controlled studies showed that the post-irradiation reduction process is kinetically suppressed in the absence of laser-generated AuNP seeds, suggesting that laser processing is important for AuNT formation. These studies represented a new approach to synthesize AuNTs without additional surfactants and extended the application of high-intensity ultrafast laser pulses for the synthesis of shape-controlled metal nanoparticles. At the end of this section, the comparisons on the reported synthetic methods of AuNTs from the perspective of the efficiency, product quality, yield, reproducibility and scalability are summarized in Table 2.

## 3. Self-Assembly of AuNTs

Self-assembly of anisotropic AuNTs into different dimensional structures has attracted great interest due to the plasmonic coupling of the edge to edge, face to face and the corner to corner of the neighboring AuNTs (Figure 4a). To achieve this, site-specific modification of AuNTs was crucial for their self-assembly, which was driven by the interaction between the surface ligands. It was reported that the edges of AuNTs showed higher reactive activity than the top/down faces [76]. Therefore, the specific modification of AuNTs could be achieved by adjusting the stoichiometric number of the ligand. The self-assembly of AuNTs into ordered monolayer and multilayer structures was achieved by modifying the nanotriangles with charged surfactants to change the repulsive electrostatic interactions [77]. The repulsive electrostatic interactions could weaken the strong van der Waals attractions between the triangular planes and effectively acted as the “molecular lubricant” that allowed particles to fine tune their mutual orientation. The morphology of the AuNT assemblies was predominantly dependent on the functionalization of the AuNTs. (1) For AuNTs stabilized by bilayer CTAB, the repulsive electrostatic interaction was smaller than the van der Waals force, resulting in rapid aggregation. (2) For thiol-terminated alkyl trimethyl ammonium chloride (TMA) ligand-functionalized AuNTs, the surface of AuNTs was covered by the monolayer of TMA instead of the bilayer structure of CTAB. In the presence of residual CTAB, the balance between attraction and repulsion controlled the self-assembly to adjust the AuNT mutual orientation during crystallization, forming an ordered multilayer structure. (3) For TMA-functionalized AuNTs with excess CTAB, the interaction between the AuNTs was repulsive, and the face-to-face stacking into multilayers was energetically unfavorable. Thus, the edge-to-edge monolayer assemblies formed. The beveled AuNTs could form polymorphic assemblies such as the honeycomb lattice and micron-sized 3D supracrystals by controlling the intricate interplay of depletion attraction and electrostatic repulsion [46]. The depletion attraction originated from non-adsorbing CTAC micelles in the suspension. Thus, the depletion attraction could be controlled by changing the concentration of CTAC in the case of micelle formation. While the electrostatic repulsion originated from the adsorbing CTAC with a positive charge, this self-assembly was dependent on the CTAC-concentration-induced depletion attraction. The planar honeycomb lattice formed under a low depletion attraction, while an interlocking honeycomb lattice and its “supracrystal” structure were assembled at a high depletion attraction (Figure 4b).

The large-area CTAB-coated AuNT monolayer at the air–water interface could be prepared by using the Langmuir–Schaefer technique [58,79]. The AuNTs were functionalized by the amphiphilic PVP, which was dispersible in organic media. The dispersed PVP-functionalized AuNTs in a mixture of ethanol/hexane were added into water. After the evaporation of organic solvents, 2D monolayers formed at the air–water interface. The uniform AuNT monolayers could be transferred to different substrates without changing the morphological integrity. Moreover, the self-assembly process was influenced by the boiling point (b.p.) and vapor pressure of the solvent. The solvent with a lower b.p. exhibited a higher evaporation speed to produce patches of AuNTs with larger gaps, while the solvent with a higher b.p. and a slower evaporation speed gave AuNTs a monolayer with a larger interprism spacing.

An external stimulus such as light could direct the reversible self-assembly of AuNTs under low-power intensities (below $0.1\;\mathrm{mW}/\mu m^{2}$) based on the plasmon-enhanced thermophoresis [80]. When the laser beam illuminated the plasmonic substrate, the photothermal effect resulted in the formation of a temperature gradient of the surrounding environment. The temperature differences generated thermophoresis, which enabled the AuNT formation at the laser spot. The AuNT assembly in a closely-contacted fashion was stabilized by the van de Waals attraction. Once the laser was turned off, the AuNTs could be redispersed into the solution. In this case, the electrostatic repulsion from the positive charge of AuNTs induced the disassembly of AuNTs. The balance among thermophoretic-induced electric force, electrostatic repulsion and van der Waals attraction induced the reversible self-assembly of AuNTs.

DNA nanotechnology provided an efficient self-assembly strategy to prepare various nanostructures. Wang and coworkers reported the directional self-assembly of DNA-modified AuNTs to form controlled assemblies [78]. The surface of AuNTs could be regio-selectively modified by using two different double-stranded DNA due to the different chemical reactivity of the edges and up/down faces in AuNTs (Figure 4c). The self-assembly of AuNTs into a face-to-face or edge-to-edge fashion was controlled by the terminal single base pairing or unpairing. Furthermore, the self-assembly process was driven by blunt-end stacking interactions on one target region of AuNTs and repulsion induced by attrition motions of terminally-mismatched nucleobases in the other region. The dsDNA-modified AuNTs preferentially self-assembled in a face-to-face manner when perfectly-matched DNA duplexes were formed on the top/bottom faces, and end-mismatched DNA duplexes were formed on the edges. When perfectly-matched DNA duplexes were formed on the edges and DNA duplexes with mismatched ends are formed on the top/bottom sides, the dsDNA-modified AuNTs selectively formed the edge-to-edge assemblies.

## 4. Applications

The dispersed anisotropic AuNTs showed a strong absorption in the near-infrared (NIR) window and good chemical stability and biocompatibility [81]. Furthermore, the assembly of AuNTs exhibited collective plasmonic properties, which were different from their individual counterparts. These properties may have broad applications in many fields such as photothermal therapy, bioimaging and sensing.

### 4.1. Biological Applications

AuNPs have broad biological applications in terms of diagnosis, drug carriers and imaging [82]. Therefore, it is essential to understand the effects of size and shape on the nanomaterials in vitro cellular uptake rate, enrichment and cytotoxicity. Therefore, the effect of NP shapes including spherical nanoparticles (SNPs) and AuNTs on the uptake of cancer cells was investigated. AuNTs and SNPs showed reversed size-dependent relationships with respect to cellular uptake. The longer side length of AuNTs showed a higher level of cellular uptake, while the larger-sized SNPs gave the lower level with regard to the cellular uptake. This phenomenon was attributed to the edges and vertices of AuNTs with a high local curvature, which accelerated their cellular internalization. The cellular uptake of nanoparticles included two processes: the adhesion of nanoparticles to the cell membrane; and the internalization of nanoparticles by the cells via energy-dependent pathways. The stronger adhesion force of the larger AuNTs might contribute to the above cell-uptake tendency. Some of the literature has reported that nanodisks with flat surfaces showed higher cell uptake than nanorods with similar surface characteristics because of the lower membrane deformation energy of the nanodisks [83,84]. With the increasing diameter of the nanodisk, the uptake level increased, perhaps due to the larger adhesion area. Ghandehari et al. demonstrated that gold nanorods (aspect ratio, 4.5) showed a longer circulation time compared with SNPs, due to the fact that macrophages take up SNPs four times more efficiently than nanorods [85]. The shape of nanoparticles is the main factor responsible for their high level of cell uptake [86,87,88]. Lin et al. reported anisotropic AuNPs with different shapes including stars, rods and triangles on cell uptake. AuNTs showed the highest efficiency of uptake by RAW264.7 cells among the above three kinds of nanoparticles [89]. Mechanistic studies revealed that different-shaped AuNTs exhibited different cell uptake mechanisms. Particularly for AuNTs, multiple endocytosis pathways including clathrin-mediated endocytosis, dynamin-dependent pathways and cytoskeletal rearrangement were involved. This might be caused by the different membrane bending energies for the nanoparticle shape effect. These data supported that AuNTs might provide an excellent platform for efficient drug delivery into cells. Inspired by the research from Ref. [82], Krishnan et al. reported the synthesis of thiol-terminated methoxy-poly (ethylene glycol) (mPEG)-modified AuNTs (pAuNTs) and their X-ray radiosensitization characteristics in vivo utilizing the intrinsic radiosensitization characteristics of AuNTs [90]. The in vitro data showed that pAuNTs could be taken up by cancer cells through endocytosis and accumulated around the nucleus and in the tumor position. However, the pAuNTs alone could not inhibit the growth of tumor. However, the X-ray irradiation induced a decrease in tumor volumes. Furthermore, the PEG-modified AuNTs exhibited low cytotoxicity and good biocompatibility. This study indicated that the pAuNTs could be used for radiosensitization in therapeutics.

The surface ligand of AuNPs played an important role in the specific recognition of target. The peptide P75 showed a specific affinity for epidermal growth factor receptor [48]. Based on this effect, the mPEG- and target peptide P75-modified AuNTs showed increased accumulation in tumor cells, and further acted as computed tomography and photoacoustic imaging-guided photothermal therapy of non-small-cell lung cancer (Figure 5a). In addition, AuNTs could be used for two-photon-induced photoluminescence, which could improve the resolution of optical microscopy imaging and effectively avoid tissue damage caused by the excitation light source [91].

The efficient synergistic antibacterial platform based on the excellent photothermal conversion efficiency of AuNTs and peroxidase-like activity was reported in [93]. The ultralow H_2_O_2_ concentration produced OH radicals by the peroxidase-like activity of AuNTs. Under low-laser power density and an ultrashort exposure time, this method protected normal tissue from damage, and could kill 99.98% of methicillin-resistant staphylococcus aureus within 3 min, which was 40% faster than photothermal healing alone.

### 4.2. Sensing

The optimization of the plasmonic effect for enhanced Raman scattering largely depends on the properties of the enhanced substrate. The anisotropy of AuNPs offers the possibility to tune their plasmonic properties and the inherent electromagnetic “hot spots”, thus becoming an effective means to improve the resolution of surface-enhanced Raman spectroscopy (SERS) [13,94].

Cytochrome c (Cyt c) is a key biomarker during the early apoptosis process [95]. However, the quantitative detection of Cyt c in living cells remains a great challenge. The AuNTs provided both SERS properties and fluorescence-quenching effects, which could be applied for detecting active substances in living biological systems. It was shown that the aptamer of Cyt c could specifically recognize Cyt c. The Cyt c aptamer and its partially-complementary sequence-modified Cy5-coated AuNTs as a nanosensor for the quantitative analysis of Cyt c in living cells was achieved using SERS-fluorescence mode. This strategy combined the advantages of the high sensitivity of SERS and the intuitive visualization of fluorescence, enabling real-time monitoring of the mitochondrial-to-cytoplasmic Cyt c translocation behavior with good sensitivity and a lower limit of detection in living cells. AuNT assemblies could also act as optical antennas to enhance multiphoton imaging capabilities in living cells [96]. Experimental data demonstrated that antibody-conjugated assemblies could show highly selective and bright multimodal second harmonic generation/two-photon fluorescence imaging of liver cancer cells. Adiponectin is a key biomarker for the early detection of gestational diabetes mellitus (GDM) [49]. Choi et al. reported a biological probe for the optical detection of adiponectin by AuNTs, whose surface was modified with thiol-terminated Raman reporter 4-mercaptobenzoic acid (4-MBA) for the detection of the antibody. In addition, 2-Mercaptoethyl ether acetic acid was modified on the surface of AuNTs as stabilizers. The carboxyl group was activated and reacted with the NH_2_ group of the biomolecules. This immune-SERS probe showed high adiponectin detection with a wide assay range, good reliability, a low limit of detection and excellent selectivity (Figure 5b).

Poly (3,4-ethylenedioxythiophene) (PEDOT) showed high electrical conductivity, biocompatibility and stability [63]. Κ-carrageenan (kC) as a sulphonated polysaccharide could form micellar system to promote the electrochemical polymerization of PEDOT. Therefore, the PEDOT/kC could be used as the coating for AuNTs to produce PEDOT/kC/AuNTs, which entrapped the acetylcholinesterase enzyme (AChE) for biosensor applications. AuNTs as electrochemical mediators for the electrochemical oxidation of acetylcholine increased the number of enzymatic products. The increased surface area of PEDOT/kC/AuNTs/AChE provided better electrical connection between electrode active sites than the bare gold electrode, thus making electron transfer easier. The voltammogram showed an increase in hysteresis, demonstrating the electrochemical activity of PEDOT/kC/AuNTs-modified electrodes. This property enabled the construction of biosensors for the selective detection of organophosphorus pesticides in real samples. AuNTs modified with mono-6-thio-β-cyclodextrin without an alkyl chain could be used as SERS substrates for trace analysis of the explosive agent 2,4-dinitrotoluene [97]. Compared with other methods, the limit of detection could be significantly improved to a sub-ppb level.

The thiol-oligonucleotide-modified AuNTs were applied to detect DNA hybridization behavior at room temperature [98,99]. When the DNA target was complementary to the probe, hybridization occurred and the amount of DNA at the surface of AuNTs increased, thus inducing an obvious red-shift of the plasmonic peak in the UV-vis spectrum, while the plasmon peak remained the same for no occurrence of hybridization. The DNA-assisted AuNTs were used for colorimetric determination of mercury ions [92]. DNA oligomers composed of poly-deoxycytidine (polyI) acted as masking agents binding to Ag(I) to prevent the interference of Ag(I) ion-induced plasmon shifts. The reduction of Hg(II) to Hg(0) by ascorbic acid resulted in amalgamation and led to its deposition on the surface of AuNTs. This led to a significant morphological transition from triangular to irregular disks, resulting in the UV-vis spectra change. With the increase in Hg(II) concentrations, the blue-shift of the plasma peak of AuNTs gradually decreased, and the solution color changed from light blue to orange-red. These results demonstrated the superior performance of amalgamation for the colorimetric determination of mercury ions. DNA-assisted AuNTs amalgamation-sensing systems could detect Hg(II) specifically in a complex solution environment compared to that of Ag nanoplates (Figure 5c).

The AuNT monolayer assembly was used for X-ray temperature calibration and SERS monitoring of plasma-driven dimerization reactions of 4-nitrothiophenol (4-NTP) to 4,4’-dimercaptoazobenzene (DMAB) [50]. The 4-NTP monolayer was modified on the AuNTs via the S−Au bond. With increasing laser power or irradiation time, the super-hot electrons generated by AuNTs increased the speed of the dimerization of 4-NTP to DMAB. The time-dependent SERS spectra showed that the AuNT monolayer effectively drove the photocatalytic reaction, so SERS could monitor the reaction kinetics in real time. Moreover, the AuNT monolayer could be used for detecting small temperature changes of AuNTs under continuous X-ray irradiation (Figure 5d).

## 5. Summary and Outlook

AuNTs with a triangular anisotropic shape showed unique optical characteristics, high electron density and dielectric properties. The physical properties of these AuNTs could be well controlled by the self-assembly approach. Many studies demonstrated that the “hot spots” of the assembled structure were generated on the edge, up/down faces and the corners, thus showing distinctively different properties from that of individual AuNTs. These broaden the practical applications of AuNTs in photothermal therapy and SERS platforms for sensing.

In summary, seed-mediated growth is the most popular method to synthesize AuNTs. However, reproducibility is difficult to guarantee due to the experimental operation of the fast-addition procedure. The controllability has been improved by adding additional stabilizers such as PVP, adjusting the temperature and pH or using a mild reducing agent such as 3-butenic acid (3BA). On the other hand, the development of the synthetic protocols of AuNTs with high quality and yields based on the nucleus-free process are still challenging. By contrast, biosynthesis is an economical and environmentally-friendly method that does not use toxic reagents such as CATB and CATC. The physical stimulus-induced synthesis is a novel method for the synthesis of shape-controlled AuNTs through high-intensity laser pulses and electric field. Nevertheless, these two methods are emerging but not commonly used, and are expected to synthesize AuNTs with good quality and scalable performance in future.

Some of the literature has reported the self-assembly of ligand-modified AuNTs into different dimensional “superstructures” through a face-to-face or edge-to-edge fashion. However, the choice of the surface ligands was still limited. The popular ligands were generally surfactants such as CTAB or CTAC and the and biological molecule DNA with a thiol termination. Therefore, in the future, more natural or artificial ligands such as small molecules and polymers with various interactions to assemble with AuNTs into superstructures will be in high demand. These assemblies are expected to be the hierarchical structure like clusters and vesicles. In addition, the characterization techniques for directly detecting the self-assembly of AuNTs in solutions has yet to be developed.

In terms of the applications of AuNTs, the surface modification using ligands with biocompatible and nontoxic properties to substitute the toxic CTAB or CTAC surfactants is essential. The high photothermal conversion, stability and biological compatibility of these AuNTs make them excellent candidates not only for applications in photothermal therapies, but also for temperature-triggered drug release and CT imaging to realize the real-time depiction of tumor boundaries. AuNTs as radiosensitizers and bioprobes for monitoring plasmon-driven catalytic reactions, detecting DNA hybridization and biomarkers for diseases have been achieved. These SERS nanosensors showed higher sensitivity, versatility and portability, but the majority of the reported applications were focused on the dispersed AuNTs or their assemblies after drying. The unique AuNT assemblies with good water dispersibility and collective physical properties for the above biological applications are expected in future.

## Figures and Tables

**Figure 1 molecules-27-08766-f001:**
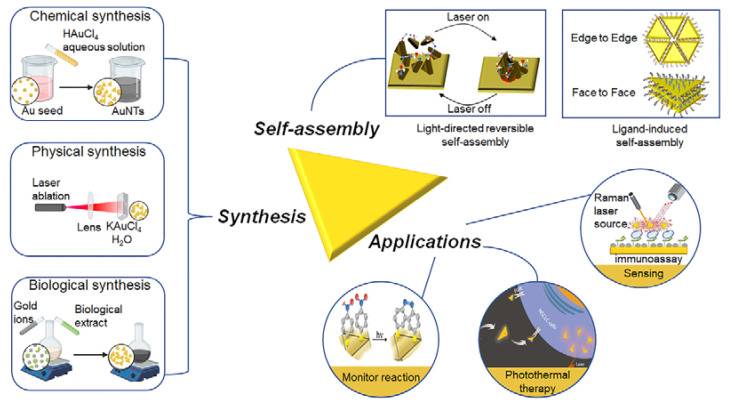
Schematic presentation of the main contents of this review. Synthesis, self-assembly of AuNTs and their applications. Taken with permissions from [48,49,50]. Adapted with permission from Ref. [48]. Copyright © 2022 American Chemical Society. Adapted with permission from Ref. [49]. Copyright © 2022 Elsevier B.V. Adapted with permission from Ref. [50]. Copyright © 2022 American Chemical Society.

**Figure 2 molecules-27-08766-f002:**
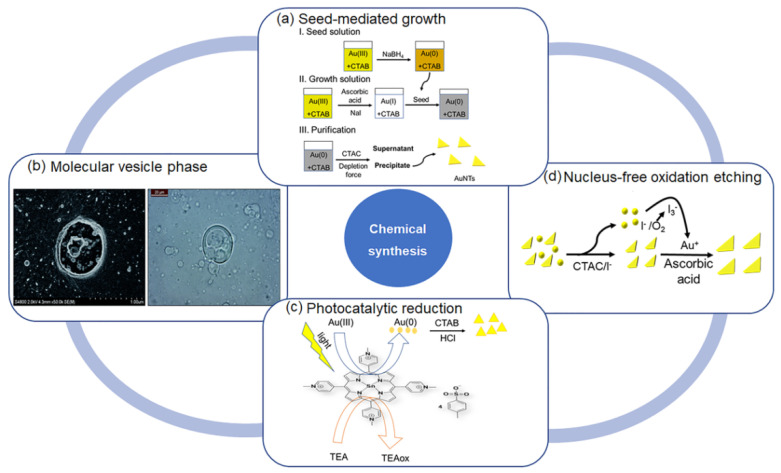
Chemical synthesis of AuNTs based on (**a**) seed-mediated growth; (**b**) molecular vesicle phase. Reprinted with permission from Ref. [61]. Copyright © 2022 The Royal Society of Chemistry; (**c**) photocatalytic reduction; and (**d**) nucleus-free oxidative etching.

**Figure 3 molecules-27-08766-f003:**
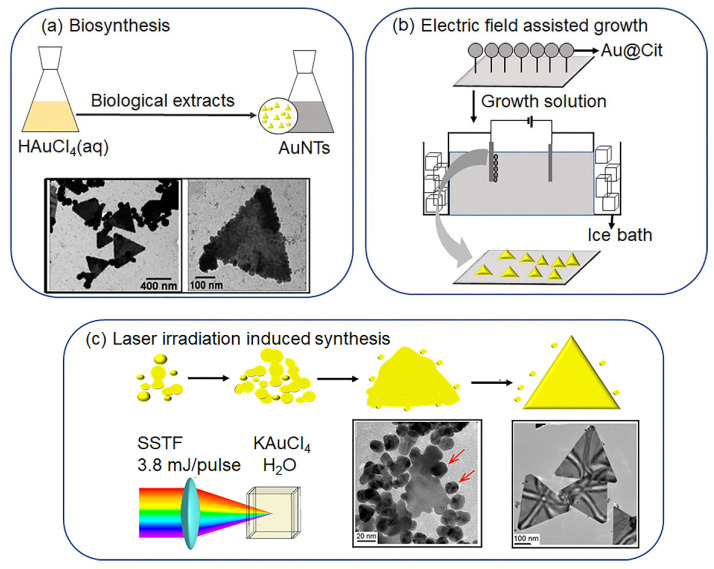
Biosynthesis and physical-stimulus-induced synthesis of AuNTs. (**a**) Biosynthesis. Reprinted with permission from Ref. [69]. Copyright © 2022 JohnWiley & Sons, Inc.; (**b**) Electric field-assisted growth; (**c**) Laser-irradiation-induced AuNTs formation. Adapted with permission from Ref. [73]. Copyright © 2022 American Chemical Society.

**Figure 4 molecules-27-08766-f004:**
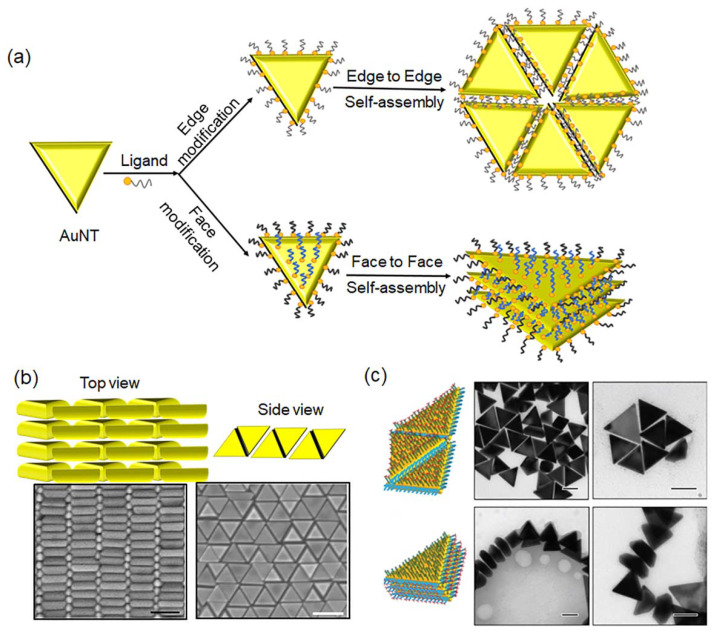
(**a**) Schematic diagram of gold nano triangle self-assembly process. (**b**) Self-assembly of gold nano triangles in the presence of CTAC micelles. Adapted with permission from Ref. [46]. Copyright © 2022 American Chemical Society; (**c**). Self-assembly of DNA-modified AuNTs into ordered structures. Reprinted with permission from Ref. [78]. Copyright © 2022 MDPI. (**b**,**c**) Scale bars are 100 nm.

**Figure 5 molecules-27-08766-f005:**
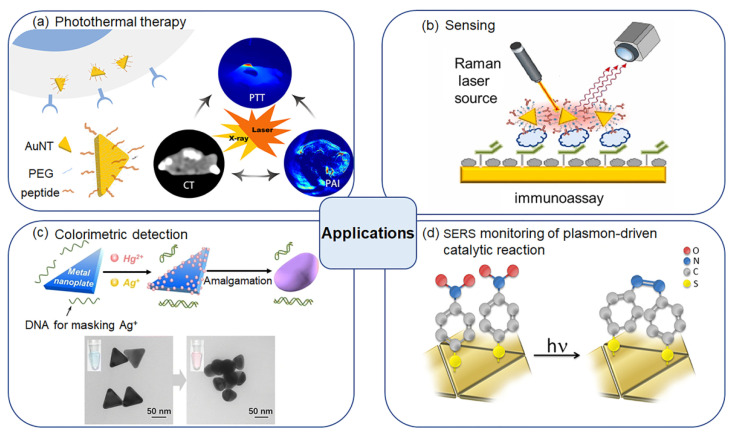
Applications of AuNTs and their assemblies. (**a**) Anti-EGFR peptide-conjugated AuNTs for computed tomography/photoacoustic imaging-guided photothermal therapy of non-small-cell lung cancer. Adapted with permission from Ref. [48]. Copyright © 2022 American Chemical Society; (**b**) AuNT assemblies for early detection of gestational diabetes mellitus. Adapted with permission from Ref. [49]. Copyright © 2022 Elsevier B.V.; (**c**) Colorimetric determination of mercury (II) ion based on DNA-assisted AuNTs. Adapted with permission from Ref. [92]. Copyright © 2022 Springer-Verlag GmbH Austria; (**d**) AuNTs for SERS monitoring of plasmon-driven catalytic reactions. Adapted with permission from Ref. [50]. Copyright © 2022 American Chemical Society.

**Table 1 molecules-27-08766-t001:** Summary of the morphology, LSPR, and synthetic methods of some reported plasmonic metal nanostructures. The pictures in Table 1 were obtained from the literature with slight modifications.

PlasmonicNanoparticles	Morphology	LSPR (nm)	Synthetic Methods
Cu spheres	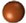	600~700	Seed-mediated growth method;Photo-chemical synthesis;Biosynthesis
Ag spheres	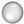	380~490
Au spheres	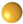	510~570
Au stars	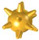	520~630	Seed-mediated growth method
Au cubes	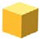	520~630	Seed-mediated growth method
Au triangles	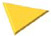	600~1300	Seed-mediated growth method;Biosynthesis;Physical stimulus induced synthesis
Au nanorods	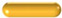	550~1100	Template synthesis; Hydrothermal synthesisSeed-mediated growth method;
Au nanocages	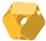	680~990	Seed-mediated growth method;Template synthesis
Au dumbbells	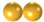	680~1090	Seed-mediated growth method
Au nanodendrites	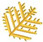	550~600	Electrochemical deposition
Au icosahedrons	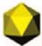	500~600	Seed-mediated growth method;Hydrothermal synthesis

**Table 2 molecules-27-08766-t002:** Comparisons on the reported synthetic methods of AuNTs regarding the efficiency, product quality, yield, reproducibility and scalability.

Synthetic Methods	Efficiency	Product Quality	Yield	Reproducibility	Scalability
Seed-mediated growth	High	Good	High	Relatively good	Wide
Molecular vesicle phase	Moderate	Good	Moderate	Good	Narrow
Photocatalysis	High	Good	High	Good	Narrow
Nucleus-free method	High	Moderate	High	Moderate	Narrow
Biosynthesis	Moderate	Moderate	Low	Low	Narrow
Physical synthesis	High	Good	Moderate	Good	Moderate

## Data Availability

Not applicable.

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
