# Peer review of "A Review on Gold Nanotriangles: Synthesis, Self-Assembly and Their Applications"

_molecules, 2022, doi:10.3390/molecules27248766_

Round 1

Reviewer 1 Report

In my opinion, the publication can be accepted after a several corrections.

In terms of the structure of the article, the following should be completed:

*  Since in the article the authors compare the methods of obtaining AuNTs, it would be useful for the reader to include a tabular comparison of the methods in terms of method effectiveness, scalability, etc.

* In the application section, the authors should comment on the comparison of AuNTs with other anisotropic structures, how in terms of physicochemical parameters (numbers) AuNTs prevail over other anisotropic gold nanoparticles in a given case

* When discussing biosynthetic methods in 2-3 sentences, it should be emphasized that the disadvantage of these methods is the lack of uniformity in the composition of raw materials of biological origin, the quality of which depends, for example, on extraction methods, etc.

* All figures, but mostly Figures 1 and 2 should be corrected for resolution and font size

* The introduction should also include a broader discussion comparing AuNTs with other anisotropic structures

In general, there are quite a few linguistic or terminological errors in the text, starting with the construction of the abstract. Here's an example:

The paper should be The abstract from 11-14 line should be corrected should be corrected to make it more accessible to the reader

13 - please dlate "practical", generally the meaning of application covers it

14 - "only few"

106 - please use efficiency instead of power

Please review the article again for corrections of similar wording and errors

I also reserve the right to draw attention to additional linguistic or terminology corrections in the next round of corrections

Author Response

Editor,
Molecules
                                                             3rd December 2022

Dear Editor,

Thank you very much for kindly reviewing our manuscript (Manuscript ID: molecules-2057218). We carefully revised our manuscript according to the valuable comments from the referees. The added sentences and references, revised linguistic or terminological errors in the text are highlighted in red in this revised manuscript. We hope that the revised manuscript now is suitable for publication in Molecules.

Point-by-point responses to the reviewers’ comments are provided below.

Sincerely yours,

Jinjian Wei

Lecture, PhD

College of Chemistry, Chemical Engineering and Materials Science,

Shandong Normal University

Jinan 250014, P.R. China

E-mail: jinjian.wei@sdnu.edu.cn
Response to the reviewers' comments

Journal: Molecules

Manuscript ID: molecules-2057218

Title: A Review on Gold Nanotriangles: Synthesis, Self-Assembly and Their Applications

Author(s): Xiaoxi Yu, Zhengkang Wang, Handan Cui, Xiaofei Wu, Wenjing Chai, Jinjian Wei*, Yuqin Chen* and Zhide Zhang*

We would like to thank the reviewers for their valuable comments and suggestions. To address these comments and suggestions we have edited and revised the initial text. Please find below our point-by-point responses and revisions according to the reviewers' comments.

Reviewer 1

General comments: In my opinion, the publication can be accepted after a several corrections. In terms of the structure of the article, the following should be completed.

Answer to general comments: Thank you very much for your positive response. We revised the manuscript point-to-point according to your valuable comments.

Comment 1: Since in the article the authors compare the methods of obtaining AuNTs, it would be useful for the reader to include a tabular comparison of the methods in terms of method effectiveness, scalability, etc.

 Answer to Comment-1: Thank you very much for your important comment. According to your comment, we added a table for the comparison of different prepared methods for AuNTs. This table included the comparisons on the various synthetic methods in terms of efficiency, product quality, yield, reproducibility, and scalability. And the newly added sentence is marked with red in Page 7 as following.  Original expressions:

These studies represented a new approach to synthesize AuNTs without additional surfactants and extended the application of ultrafast laser pulses with high-intensity for the synthesis of shape-controlled metal nanoparticles.

Revised expressions:

These studies represented a new approach to synthesize AuNTs without additional surfactants and extended the application of ultrafast laser pulses with high-intensity for the synthesis of shape-controlled metal nanoparticles. At the end of this section, the comparisons on the reported synthetic methods of AuNTs from the perspective of the efficiency, product quality, yield, reproducibility, and scalability were summarized in Table 2.

Table 2. Comparisons on the reported synthetic methods of AuNTs regarding the efficiency, product quality, yield, reproducibility, and scalability

      Comment 2: In the application section, the authors should comment on the comparison of AuNTs with other anisotropic structures, how in terms of physicochemical parameters (numbers) AuNTs prevail over other anisotropic gold nanoparticles in a given case. Answer to Comment-2: Thank you very much for your important comment. According to your comment, we discussed the effect of nanoparticle shape on the cell uptake. And we also included the physicochemical parameters of AuNTs prevailing over other anisotropic gold nanoparticles in a given case. Original expressions:AuNPs have broad biological applications in terms of diagnosis, drug carriers and imaging [12]. Therefore, it is essential to understand the effect of size and shape of the nanomaterials in vitro cellular uptake rate and enrichment, and cytotoxicity. Therefore, the effect of NP shapes including spherical nanoparticles (SNPs) and TNPs on the uptake of cancer cells was investigated. TNPs and SNPs showed reversed size-dependent relationships with respect to cellular uptake. The longer side length of TNPs showed a higher level of cellular uptake, while the larger sized-SNPs gave lower level with regard to the cellular uptake. This phenomenon was attributed to the edges and vertices of TNPs with high local curvature, which accelerated their cellular internalization. These data supported that AuNTs might provide an excellent platform for efficient drug delivery into cells. Revised expressions:AuNPs have broad biological applications in terms of diagnosis, drug carriers and imaging [79]. Therefore, it is essential to understand the effects of size and shape on the nanomaterials in vitro cellular uptake rate, enrichment, and cytotoxicity. Therefore, the effect of NP shapes including spherical nanoparticles (SNPs) and AuNTs on the uptake of cancer cells was investigated. AuNTs and SNPs showed reversed size-dependent relationships with respect to cellular uptake. The longer side length of AuNTs showed a higher level of cellular uptake, while the larger sized-SNPs gave the lower level with regard to the cellular uptake. This phenomenon was attributed to the edges and vertices of AuNTs with high local curvature, which accelerated their cellular internalization. The cellular uptake of nanoparticles included two processes:Adhesion of nanoparticles to the cell membrane and internalization of nanoparticles by the cells via energy-dependent pathways. The stronger adhesion force of the larger AuNTs might contribute to the above cell-uptake tendency. Some literatures reported that nanodisks with flat surfaces showed higher cell uptake than nanorods with the similar surface characteristics because of the lower membrane deformation energy of nanodisks [80,81]. With increasing the diameter of nanodisk, the uptake level was increased probably due to the larger adhesion area. Ghandehari et al. demonstrated that gold nanorods (aspect ratio, 4.5) showed a longer circulation time compared with SNPs, due to the fact that macrophages uptake SNPs four times more efficiently than nanorods [82]. The shape of nanoparticles is the main factor for the high level of cell uptake [83-85]. Lin et al. reported anisotropic AuNPs with different shapes including stars, rods, and triangles on cell uptake. AuNTs showed the highest efficiency of uptake by RAW264.7 cells among the above three kinds of nanoparticles [86]. Mechanistic studies revealed that different-shaped AuNTs exhibited different cell uptake mechanisms. Particularly for AuNTs, multiple endocytosis pathways including clathrin-mediated endocytosis, dynamin-dependent pathway and cytoskeletal rearrangement were involved. This might be caused by the different membrane bending energies for the nanoparticle shape effect. These data supported that AuNTs might provide an excellent platform for efficient drug delivery into cells. Comment 3: When discussing biosynthetic methods in 2-3 sentences, it should be emphasized that the disadvantage of these methods is the lack of uniformity in the composition of raw materials of biological origin, the quality of which depends, for example, on extraction methods, etc. Answer to Comment-3: Thank you very much for your important comment. According to your comment, we illustrated the disadvantages of biosynthesis in Page 7 as following. However, the quality of the biological raw materials might not be uniform in various regions or countries. Further, the extraction conditions such as the extraction solvent and temperature largely influenced the purity of the active target substances. These factors could affect the efficiency and morphology control of AuNT synthesis. Comment 4: All figures, but mostly Figures 1 and 2 should be corrected for resolution and font size.Answer to Comment-4: Thank you very much for your important comment. We changed the resolution and font size of all figures as required. And the newly revised figures instead of the old ones are inserted in the main text. Comment 5: The introduction should also include a broader discussion comparing AuNTs with other anisotropic structures.Answer to Comment-5: Thank you very much for your important comment. We added a broader discussion of other anisotropic nanostructures in the introduction as follows. Original expressions:And the LSPR band was largely influenced by the shape and size of AuNPs [7-8]. Among the various shaped AuNPs, gold nanotriangles (AuNTs) with unique triangular anisotropy exhibited special optical properties, in which the three sharp vertices exhibited a significant hot spot enhancement effect. Revised expressions:Different from the isotropic nanoparticles, anisotropic nanoparticles such as nanorods and nanostars exhibited disparate plasmonic resonance modes, thus resulting in the unique optical characteristics [30-32]. Gold nanorods showed transversal and longitudinal resonances, resulting in two typical LSPR bands [33-35]. The plasmonic peaks were largely influenced by the length, width, and aspect ratio between them. Gold nanostars are another complex nanostructures composed of inner core and outer spikes [36,37]. The length of the spikes dramatically influenced the absorption by nanostars. With increasing the length of spikes, the absorption was red-shifted. Among the various shaped AuNPs, gold nanotriangles (AuNTs) with unique triangular anisotropy exhibited special optical properties, in which the three sharp vertices exhibited a significant hot spot enhancement effect. Besides, AuNTs also showed wide and controllable LSPR by changing their sizes compared with other shaped AuNPs (Table 1). Comment 6: In general, there are quite a few linguistic or terminological errors in the text, starting with the construction of the abstract. Here's an example: The paper should be the abstract from 11-14 line should be corrected should be corrected to make it more accessible to the reader.13 - please dilate "practical", generally the meaning of application covers it14 - "only few"106 - please use efficiency instead of power Answer to Comment-6: Thank you very much for your important comment. According to your comment, we corrected the abstract from 11-14 line as following. After careful checking the manuscript, we corrected the similar linguistic or terminological errors and all the corrections were marked with red in the text. Original expressions:Compared with the spherical and rod-shaped gold nanoparticles, gold nanotriangles (AuNTs) exhibited unique optical and plasmonic properties due to their special triangular anisotropy. Indeed, AuNTs showed promising practical applications in optoelectronics, optical sensing, imaging and other fields. Revised expressions:Gold nanoparticles (AuNPs) with interesting optical properties have attracted much attention in recent years. Up to now, the synthesis and plasmonic properties of AuNPs with controllable size and shape were extensively investigated. Among these AuNPs, gold nanotriangles (AuNTs) exhibited unique optical and plasmonic properties due to their special triangular anisotropy. Indeed, AuNTs showed promising applications in optoelectronics, optical sensing, imaging and other fields. 
Reviewer: 2Comment 1: Despite the fact that the review is devoted to the triangular gold nanostructures, the authors should consider other geometric modifications of gold nanostructures and other plasmonic-active metals (silver and copper). For example, the review lacks information about such morphologies of plasmonic nanostructures as nanocubes, ellipsoidal nanoparticles, dendrites, "sunflower-like" structures, and other fractal nanostructures. The review will be significantly enriched and useful to researchers in the field of plasmonics if there is a table that lists the type of metal of the nanostructure, its shape, the method of obtaining, as well as the regions of plasmon resonance. Based on the results of the analysis of this table, it will be easier for the reader to identify the features of gold nanotriangles in comparison with other plasmonic nanostructures.Answer to Comment-1: Thank you very much for your important comment. We revised the manuscript according to your comments in the introduction as follows. Original expressions:Gold nanoparticles (AuNPs) with optical properties have attracted much attention in recent years due to their potential applications in sensing, optical materials, catalysis and biological fields [1-6]. Their optical properties originated from the localized surface plasmon resonance (LSPR). And the LSPR band was largely influenced by the shape and size of AuNPs [7-8]. Among the various shaped AuNPs, gold nanotriangles (AuNTs) with unique triangular anisotropy exhibited special optical properties, in which the three sharp vertices exhibited a significant hot spot enhancement effect. Revised expressions:

Plasmonic metallic nanoparticles such as gold [1-4], silver [5-8], and copper [9,10] showed interesting optical properties, which are originated from the collective and coherent oscillations of conductive electrons under the irradiation of light [11]. The resulting resonance band from localized surface plasmon resonance (LSPR) is largely influenced by the composition, shape and size of nanoparticles [12,13]. Several comprehensive reviews introduced the synthesis and plasmonic properties of the metallic nanoparticles with different shapes such as spheres, nanocages, rods, nanomatryoshkas, stars, dendrites, icosahedrons, dumbbells and dimmers [13-24]. Herein, we briefly summarized the classifications, plasmonic bands, and synthetic methods of different nanoparticles in Table 1. Compared with silver and copper nanoparticles, gold nanoparticles (AuNPs) with good biocompatibility and stability have attracted much attention in recent years due to their potential applications in sensing, optical materials, catalysis and biological fields [25-30]. Different from the isotropic nanoparticles, anisotropic nanoparticles such as nanorods and nanostars exhibited disparate plasmonic resonance modes, thus resulting in the unique optical characteristics [30-32]. Gold nanorods showed transversal and longitudinal resonances, resulting in two typical LSPR bands [33-35]. The plasmonic peaks were largely influenced by the length, width, and aspect ratio between them. Gold nanostars are another complex nanostructures composed of inner core and outer spikes [36,37]. The length of the spikes dramatically influenced the absorption by nanostars. With increasing the length of spikes, the absorption was red-shifted.

Table 1. Summary of the morphology, LSPR, and synthetic methods of some reported plasmonic metal nanostructures. The pictures in Table 1 were obtained from the literatures with slight modification.

Among the various shaped AuNPs, gold nanotriangles (AuNTs) with unique triangular anisotropy exhibited special optical properties, in which the three sharp vertices exhibited a significant hot spot enhancement effect. Besides, AuNTs also showed wide and controllable LSPR by changing their sizes compared with other shaped AuNPs (Table 1). Comment 2: The review has lacks information on theoretical calculations and modeling of plasmonic properties. This information should be added, because without it the review of gold nanotriangles would be incomplete. Usually, this information can be found in articles devoted to SERS.Answer to Comment-2: Thank you very much for your important comment. We added the information on theoretical calculations and modeling of plasmonic properties in the main test as following.  Original expressions:Experimental and theoretical studies revealed that the tips gave high electromagnetic field enhancement from a dipolar plasmon mode, while other modes at higher energies originated from the highest enhancement at the edges and at the center of the AuNTs [11,12]. During the past decades, many efforts were devoted to the synthetic exploration of AuNTs with high quality and obtained yields. Revised expressions:Electron energy-loss spectroscopy (EELS) experimental data of AuNTs supported that the tips gave high electromagnetic field enhancement from a dipolar plasmon mode, while other modes at higher energies originated from the highest enhancement at the edges and at the center of the AuNTs [40]. The boundary-element method (BEM) simulation supported existence of the in-plane dipole and quadrupole plasmon modes in AuNT and the EELS mapping at the corresponding energies [13,40]. The energy of plasmon mode at the corner was lower than those at the edge and face. With increasing the AuNT thickness, the energy of corner decreased, while the edge and face increased. This energy shift was caused by an increase in the free electron density in the AuNT [39]. Theoretical calculations indicated that EELS and cathodoluminescence were closely related to the optical extinction and scattering of AuNTs respectively. The dipolar mode could induce scattering blue-shifted in connection with its corresponding extinction. The magnitude of extinction and scattering increased with increasing the dissipation level [40]. During the past decades, many efforts were devoted to the synthetic exploration of AuNTs with high quality and obtained yields.

Reviewer 2 Report

In my personal opinion the introduction for review is very poor.

Despite the fact that the review is devoted to the triangular gold nanostructures, the authors should consider other geometric modifications of gold nanostructures and other plasmonic-active metals (silver and copper). For example, the review lacks information about such morphologies of plasmonic nanostructures as nanocubes, ellipsoidal nanoparticles, dendrites, "sunflower-like" structures, and other fractal nanostructures.

The review will be significantly enriched and useful to researchers in the field of plasmonics if there is a table that lists the type of metal of the nanostructure, its shape, the method of obtaining, as well as the regions of plasmon resonance.

Based on the results of the analysis of this table, it will be easier for the reader to identify the features of gold nanotriangles in comparison with other plasmonic nanostructures.

The review has lacks information on theoretical calculations and modeling of plasmonic properties. This information should be added, because without it the review of gold nanotriangles would be incomplete. Usually this information can be found in articles devoted to SERS.

Despite the weak introduction and the absence of any data on theoretical calculations and simulations of the plasmonic properties of the considered gold nanostructures, I believe that after improvements this work can be published in the Molecules.

Author Response

(The authors gave the same response as above.)
